# Self-Assembly Nanoparticles of Natural Bioactive Abietane Diterpenes

**DOI:** 10.3390/ijms221910210

**Published:** 2021-09-22

**Authors:** Epole Ntungwe, Eva María Domínguez-Martín, Gabrielle Bangay, Catarina Garcia, Iris Guerreiro, Eleonora Colombo, Lucilia Saraiva, Ana María Díaz-Lanza, Andreia Rosatella, Marta M. Alves, Catarina Pinto Reis, Daniele Passarella, Patricia Rijo

**Affiliations:** 1CBIOS—Universidade Lusófona’s Research Center for Biosciences & Health Technologies, Campo Grande 376, 1749-024 Lisbon, Portugal; epole.ntungwe@ulusofona.pt (E.N.); evam.dominguez@uah.es (E.M.D.-M.); gabrielle.bangay@gmail.com (G.B.); catarina.g.garcia@gmail.com (C.G.); iris.c.f.guerreiro@gmail.com (I.G.); andreia.rosatella@ulusofona.pt (A.R.); 2Pharmacology Area (Pharmacognosy Laboratory), New Antitumor Compounds: Toxic Action on Leukemia Cells Research Group, Department of Biomedical Sciences, Faculty of Pharmacy, University of Alcalá de Henares, Ctra. A2, Km 33.100–Campus Universitario, 28805 Alcalá de Henares, Spain; ana.diaz@uah.es; 3Department of Chemistry, University of Milan, Via Golgi 19, 20133 Milano, Italy; eleonora.colombo@unimi.it (E.C.); daniele.passarella@unimi.it (D.P.); 4LAQV-Faculty of Pharmacy of University of Porto, Rua de Jorge Viterbo Ferreira 228, 4050-313 Porto, Portugal; lucilia.saraiva@ff.up.pt; 5iMed.ULisboa, Faculdade de Farmácia da Universidade de Lisboa, Av. Prof. Gama Pinto, 1649-003 Lisboa, Portugal; catarinareis@ff.ulisboa.pt; 6Centro de Química Estrutural, Instituto Superior Técnico, Universidade de Lisboa, Av. Rovisco Pais, 1, 1049-001 Lisboa, Portugal; martamalves@tecnico.ulisboa.pt

**Keywords:** royleanones, self-assembly, nanoparticles, squalene, oleic acid, 1-bromododecane

## Abstract

Different approaches have been reported to enhance penetration of small drugs through physiological barriers; among them is the self-assembly drug conjugates preparation that shows to be a promising approach to improve activity and penetration, as well as to reduce side effects. In recent years, the use of drug-conjugates, usually obtained by covalent coupling of a drug with biocompatible lipid moieties to form nanoparticles, has gained considerable attention. Natural products isolated from plants have been a successful source of potential drug leads with unique structural diversity. In the present work three molecules derived from natural products were employed as lead molecules for the synthesis of self-assembled nanoparticles. The first molecule is the cytotoxic royleanone 7α-acetoxy-6β-hydroxyroyleanone (Roy, **1**) that has been isolated from hairy coleus (*Plectranthus hadiensis* (Forssk.) Schweinf). ex Sprenger leaves in a large amount. This royleanone, its hemisynthetic derivative 7α-acetoxy-6β-hydroxy-12-benzoyloxyroyleanone (12BzRoy, **2**) and 6,7-dehydroroyleanone (DHR, **3**), isolated from the essential oil of thicket coleus (*P. madagascariensis* (Pers.) Benth.) were employed in this study. The royleanones were conjugated with squalene (sq), oleic acid (OA), and/or 1-bromododecane (BD) self-assembly inducers. Roy-OA, DHR-sq, and 12BzRoy-sq conjugates were successfully synthesized and characterized. The cytotoxic effect of DHR-sq was previously assessed on three human cell lines: NCI-H460 (IC_50_ 74.0 ± 2.2 µM), NCI-H460/R (IC_50_ 147.3 ± 3.7 µM), and MRC-5 (IC_50_ 127.3 ± 7.3 µM), and in this work Roy-OA NPs was assayed against Vero-E6 cells at different concentrations (0.05, 0.1, and 0.2 mg/mL). The cytotoxicity of DHR-sq NPs was lower when compared with DHR alone in these cell lines: NCI-H460 (IC_50_ 10.3 ± 0.5 µM), NCI-H460/R (IC_50_ 10.6 ± 0.4 µM), and MRC-5 (IC_50_16.9 ± 0.5 µM). The same results were observed with Roy-OA NPs against Vero-E6 cells as was found to be less cytotoxic than Roy alone in all the concentrations tested. From the obtained DLS results, 12BzRoy-sq assemblies were not in the nano range, although Roy-OA NP assemblies show a promising size (509.33 nm), Pdl (0.249), zeta potential (−46.2 mV), and spherical morphology from SEM. In addition, these NPs had a low release of Roy at physiological pH 7.4 after 24 h. These results suggest the nano assemblies can act as prodrugs for the release of cytotoxic lead molecules.

## 1. Introduction

Many established and newly developed therapeutics are poorly water-soluble, limiting their bioavailability [1] and adequate drug delivery. Nanoparticles (NPs) have recently gained an increase in interest in the scientific community due to their advantages in increasing and effectively overcoming some pharmacokinetic limitations, such as drug bioavailability and possible side effects due to targeted delivery to specific tissues [2]. Important desired characteristics of nanoparticles are a large loading capacity for one or more therapeutics, a long circulation half-life, protection of the therapeutic agent during circulation, limited side effects, and an effective and selective release of therapeutics at the target site [2,3]. Several nanoplatforms already exist. These include polymer-based nanoparticles, lipid-based NPs, and nanosuspensions [4]. Lipid-based NPs are promising due to their high biocompatibility, ease of production, and solubilization capacity retention upon administration [1]. Lipid-based NPs can be divided into three different groups: liposomes, solid lipid NPs, and self-nano-emulsifying drug delivery systems (SNEDDS). The last one has received more attention from researchers due to its easy formulation. They consist of amphiphilic molecules with a lipid and a surfactant phase that, in an aqueous medium, mimic the self-assembly behavior that occurs in nature [4,5], forming highly ordered structures. This spontaneous assembly occurs due to non-covalent interactions such as van der Waals interactions, hydrogen bonding, and electrostatic interactions. Thus, it is possible to obtain fine and stable nanoemulsions upon gentle agitation. This is especially interesting for industrialization and commercialization because no extra energy needs to be added [4]. By using self-assembling conjugates it is possible to increase the loading capacity of the NPs and eliminate the need of a carrier [6]. This overcomes important drawbacks of nanomedicines including the use of many excipients, low drug loading efficiency, and crystallization [7,8].

To form self-assembling NPs, the conjugate is usually formed by covalent coupling of the drug with biocompatible lipide moieties [9]. These molecules can be connected through smart linkers, which release the drug when an internal or external stimulus is applied, ensuring targeted delivery and controlled drug release [10]. A multitude of smart linkers (hydrazone, azo, peptide, disulfide, etc.) are available, each with unique advantages and disadvantages [11], and the most frequently used are redox-sensitive linkers. For instance, the high concentration of glutathione (GSH) in tumor cells cleaves these linkers, and the drug is released. The use of these types of linkers has been demonstrated for NPs of multiple cytotoxic agents [12]. Squalene is a lipidic moiety and a natural precursor of many steroids that spontaneously form self-assembling NPs in water, with an unusually high drug content. Since squalene is well tolerated, it makes it an ideal component of the drug delivery system [7,9,13]. Another attractive option commonly used as lipidic moieties is fatty acids due to their biodegradability and biosafety. Previous studies have concluded that oleic acid and 1-bromododecane can improve the stability of anticancer drugs and accomplish targeted delivery of the drug [6,14,15].

*Plectranthus* plants have historically been used in traditional medicine and are a known source of bioactive products, particularly royleanones, with pharmacological activities, including antiproliferative properties [11,16]. On one hand, 7α-acetoxy-6β-hydroxyroyleanone (Roy), for example, is the main metabolite of *P. hadiensis* leaf extracts [17], and it has been shown to have cytotoxicity against different cancer cell lines. It regulates MDR by inhibiting P-glycoprotein [18,19]; thus, is an interesting component for the development of new cancer treatments. The presence of hydroxyl groups at positions 6 and 12 of this cytotoxic molecule makes it an attractive lead for derivatization for the drug discovery process. 7α-Acetoxy-6β-hydroxy-12-benzoyloxyroyleanone (12BzRoy) is a derivative of Roy obtained through a benzoylation reaction and has shown improved cytotoxic activity in different cancer cell lines. On the other hand, 6,7-dehydroroyleanone (DHR) is the main component of *P. madagascariensis* essential oil and was isolated from this plant, and it has been reported to have antioxidant, antimicrobial, and cytotoxic activities [11,20]. 

In this study the synthesis of lipid-drug conjugates based on three abietane diterpenes (Roy, its derivative 12BzRoy, and DHR, Figure 1) is described, which self-assemble in water, in a way to increase the bioavailability. The NPs of these lipid-drug conjugates were characterized, and their properties as well as their cytotoxicity in different cancer cell lines were accessed.

## 2. Results and Discussion

### 2.1. Hemisynthesis of Self-Assembly Nanoparticles Conjugates

Hemisyntheses were performed with Roy, 12BzRoy, and DHR lead molecules using squalene (Sq), oleic acid (OA), and 1-bromododecane (BD) as self-assembly inducers. Unfortunately, only two conjugates were successfully synthetized: 12BzRoy-Sq (**7**) and Roy-OA conjugates (**5**), with overall yields of 20.4% and 90.9%, respectively (Figure 2). 

The NMR analysis showed that, in the case of 12BzRoy, squalene was added to the C-6 hydroxyl group (6-OH), that was confirmed by the absence of the 6-OH around 7 ppm in the ^1^H-NMR spectra Appendix A. In addition, the slight chemical shift of 7β-H from 5.70 to 5.30 and H-6α from 4.34 to 4.22 also confirmed the presence of the squalene moiety at position 6 of 12BzRoy molecule. For the Roy-OA conjugate, the presence of 6-OH and no change in the chemical shift of 7β-H (5.64 ppm) and 6α-H (4.31 ppm) of the lead molecule Roy suggests that oleic acid was added on position 12. This is in agreement with our previous work on derivatization of Roy, where position 12 was proven to be more active [11]. In addition, the number of protons of the oleic acid ethylene group in the synthesized product is proportional to one molecule of Roy, indicating the presence of one oleic acid molecule in the conjugate at position 12. DHR-Sq (**9**) structure was confirmed by NMR (Appendix A) and was in agreement with the literature [21].

### 2.2. Nanoassemblies: Preparation and Characterization

After the synthesis of the Roy and 12BzRoy conjugates, the respective self-assembled nanoparticles (NP) were prepared. In a way to confirm the formation of the NPs, FTIR spectra main peaks and the corresponding functional groups were identified for all samples tested (drug conjugate and its corresponding self-assembled NP). In the IR spectrum of compound **5** (Roy-OA conjugate), characteristic absorption bands for a hydroxyl group (3454 cm^−1^) at the 6 position, C–H alkane stretching (2923 cm^−1^), and ester groups (1725 cm^−1^) at positions 12 and 7 were evident. The peak at 1667 cm^−1^ was attributed to the carbonyl groups and at 1459 cm^−1^ to the methyl groups. In the Roy-OA self-assembled nanoparticle, there was a clear difference in the spectrum. The peak at 2923 cm^−1^ due to C–H alkane stretching disappeared, suggesting some structure modification of Roy-OA conjugate in an organized nanoparticle arrangement (Figure 3). In addition, the carbonyl region (around 1600 cm^−1^) in the Roy-OA self-assembled NP was also different from its conjugate. These results were similar to the FTIR analysis of 12BzRoy-Sq self-assembled NP when compared with the 12BzRoy-Sq conjugate. For example, displayed bands at 2924 cm^−1^ attributed to C–H alkane stretching and 1643 cm^−1^ for carbonyl groups (Figure 4) were not seen in the spectrum of 12BzRoy-Sq NP, further confirming the formation of the NPs. 

The mean size value, polydispersity, and zeta potential of the NPs were determined. In terms of size, it was observed that 12BzRoy-Sq was bigger than the Roy-OA NPs. The size of the 12BzRoy-Sq NPs was different from that previously observed for squalene-based heteronanoparticles [7,9], indicating that the larger nature of the 12BzRoy-Sq NPs could be due to the agglomeration in water; hence, these are not suitable for drug delivery. The zeta potential was negative in both cases, with the Roy-OA being more negative. This observation implies a better stability of these NPs. The Pdl of Roy-OA was smaller, indicating a narrower and homogenous size distribution (Table 1). Since the 12BzRoy-Sq NPs size was not in the nano range, only Roy-OA NPs were further characterized. 

The morphology for successful fabrication of self-assembled NPs was further confirmed by SEM. Observations showed that NPs in the Roy-OA assembly had a round shape morphology with homogenous sizes ranging from 80 to 50 nm in diameter (Figure 5b). 

### 2.3. In Vitro Release Studies 

The evaluation of Roy release from the self-assembled NPs in PBS at physiologic pH 7.4 was used to establish the release profile of Roy from the Roy-OA NP. As illustrated in Figure 6, Roy showed a slow release, since after 24 h approximately 8.35 % was released from the NPs. The release profile was continually sustained over the first 24 h of the assay; however, a bulk amount of the drug was not released from the NP, and Roy was found to degrade over time. Importantly, the amount of Roy released is enough to exhibit cytotoxic activity against cancer cell lines (Figure 7 and Figure 8). These results were in agreement with those of Luo et al., where oleic acid conjugated to Paclitaxel (PTX) was used as control. The results showed that there was almost no PTX released from PTX-OA NPs after incubation in PBS (pH 7.4) [15]. 

### 2.4. Biological Activity Study

#### 2.4.1. Preliminary Toxicity Assay

The general toxicity of Roy-OA NP and its lead molecule was assessed using the *Artemia salina* model. The results revealed that Roy-OA NP had no toxicity against this model and was eight times less toxic than the corresponding lead molecule, Roy. These suggest that the Roy-OA NP may act as a prodrug. 

#### 2.4.2. Cytotoxicity Study

Roy-OA nano assembly did not decrease cell viability of Vero-E6 cells even at high concentration of 200 µg/mL as compared to Roy, where a significant reduction in cell viability in a concentration-dependent manner was observed (Figure 8). These results are consistent with those obtained from the royleanone DHR and DHR-Sq NPs previously tested against three human cancer cell lines, namely, sensitive non-small cell lung carcinoma, NCI-H460 cell line; multidrug-resistant non-small cell lung carcinoma cell line with P-glycoprotein overexpression, NCI-H460/R; and human embryonal bronchial epithelial (MCR-5) cells. DHR was more cytotoxic: NCI-H460 (IC_50_ 10.3 ± 0.5 µM), NCI-H460/R (IC_50_ 10.6 ± 0.4 µM), MRC-5 (16.9 ± 0.5 µM) than DHR-sq NPs: NCI-H460 (IC_50_ 74.0 ± 2.2 µM), NCI-H460/R (IC_50_ 147.3 ± 3.7 µM), and MRC-5 (IC_50_ 127.3 ± 7.3 µM) [21]. This suggests that Roy-OA nano assembly may act as a strategy to vehiculate the cytotoxic Roy. This type of strategy can allow the release of the parent drug in vivo, enhancing the stability and selectivity of targeted cells, eliminating the side effects, and overcomes discomfort scents of drugs. In addition, it increases drug loading efficiency and avoids the large use of excipients [8].

## 3. Materials and Methods

### 3.1. Plant Material

The plant material (Figure 9), *Plectranthus madagascarensis* Benth. and *P. hadiensis* (Forssk.) Schweinf. ex Sprenger, was cultivated in Parque Botânico da Tapada da Ajuda (Instituto Superior de Agronomia, Lisbon, Portugal) from cuttings obtained from the Kirstenbosch National Botanical Garden (Cape Town, South Africa). Whole plants were collected between 2007 and 2008, always in June and September. They were deposited in the Herbarium “João de Carvalho e Vasconcellos” of the Instituto Superior de Agronomia, Lisboa (LISI), Portugal under the Voucher number 841/2007 for *P. madagascarensis* Benth and 833/2007 and 438/2010/ for *P. hadiensis*. The plant names were verified with the Plant List [22].

The extraction and isolation processes of **1** and **3** were performed according to Ntungwe et al., 2021 [16], and Garcia C. et al., 2018 [20], respectively. Briefly, 46.86 ± 3.51 g of dried *P. madagascariensis* was submerged in 1 L of distilled water and then submitted to hydrodistillations using a Clevenger apparatus (steam distillation) to obtain the essential oils (EOS). *P. hadiensis* extract was obtained by the ultra-sonication method, adding 30 mL of acetone to 3 g of ground dry plants sonicated for 1 h, and filtered (Whatman No 5 paper, Inc., Clifton, NJ, USA). All extractions were performed in triplicate, and the solvent was further evaporated to dryness at 40 °C on a rotary evaporator (Sigma-Aldrich, IKA HBR 4 basic heating bath, Essen, Germany).

The EOs from *P. madagascariensis* were fractionated by dry-column flash chromatography on high-purity-grade silica gel 60 (1.09385.1000; 230–400 mesh; Merck KGaA 64271 Darmstadt, Germany), with eluent of increasing polarity of *n*-hexane–EtOAc (9:1). DHR was purified by recrystallization with methanol. *P. hadiensis* extract was fractionated by normal phase column chromatography over silica gel using mixtures of *n*-hexane–EtOAc (8:2) as eluents. Roy was obtained using preparative thin-layer chromatography (*n*-hexane/AcOEt 7:3) on pre-coated TLC sheets (Merck 7747, Darmstadt, Germany).

### 3.2. Reaction Procedure

In the hemisynthetic study, the derivative (2) 7α-acetoxy-6β-hydroxy-12-benzoyloxyroyleanone (12BzRoy) was successfully prepared from the lead molecule **1** obtained on the more reactive 12-OH by benzoylation reaction (Figure 1). The reactions, purification, identification, and stability of royleanone derivatives were carried out and previously reported in the study of Garcia et al., 2020 [11].

### 3.3. Hemisynthesis of Drug Conjugates for Self-Assembly Nanoparticles

7α-Acetoxy-6β-hydroxyroyleanone (1), 7α-acetoxy-6β-hydroxy-12-benzoyloxyroyleanone (2), and 6,7-dehydroroyleanone (3) were used as lead molecules in this study to prepare conjugate compounds using squalene, oleic acid, and 1-bromododecane as self-assembly inducers exploiting different methodologies.

#### 3.3.1. Synthesis of Drug Conjugates Using Squalene as a Self-Assembly Inducer [7]

For the general procedure, 1-ethyl-3-(3-dimethylaminopropyl) carbodiimide hydrochloride (EDC·HCl, 0.06 mmol, VWR) and 4-dimethylaminopyridine (DMAP, 0.03 mmol, VWR) were added to a solution of squalene linker (0.04 mmol), in dry CH_2_Cl_2_ (1 mL, Sigma). The lead molecule (0.04 mmol) was added, and the reaction mixture was stirred at room temperature for 3–96 h. HCl 1 M (15 mL) was added and extracted with CH_2_Cl_2_ (5 × 5 mL). The organic layers were dried over Na_2_SO_4_, and the solvent was removed under reduced pressure. The crude was purified by semi-preparative TLC (AcOEt/Hex). (Figure 2 and Figure 3).

#### 3.3.2. Synthesis of Drug Conjugates Using Oleic Acid as a Self-Assembly Inducer [6]

The lead molecule (82 mmol, 1 eq) was dissolved in anhydrous methylene chloride (1 mL), and then oleic acid (82 mmol, 1 eq), DMAP (1 eq, 82 mmol, VWR), and dicyclohexylcarbodiimide (DCC, 2 eq 164 mmol, VWR) were added. The reaction mixture was stirred at ambient temperature for 12 h. After filtering, the filtrate was diluted with diethyl ether, and then the mixture was washed with 5% aqueous hydrochloric acid, water, and saturated aqueous sodium chloride. The filtrate was evaporated, and then the residue was thoroughly dried under a vacuum to get the conjugate. (Figure 2 and Figure 3).

#### 3.3.3. Synthesis of Roy-Dodecane Conjugate Using 1-Bromododecane as Self-Assembly Inducer [14]

A bromoalkane was coupled to Roy by SN_2_ substitution. NaH (60% in mineral oil, 1.068 mg, 3 Eq, Sigma) was suspended in dry THF (1.5 mL, Sigma) in a three-necked flask, at 0 °C, and under argon atmosphere. A solution of Roy (10 mg, 1 eq) in dry THF was added dropwise, keeping the temperature below 0 °C. The reaction mixture was stirred for 30 min at 0 °C and then for 20 min 25 at ambient temperature. After cooling again to 0 °C, a solution of 1-bromododecane (17.82 µL, 3 eq) in dry THF (1 mL) was slowly added. The mixture was stirred at ambient temperature overnight, heated on steam. (Figure 2).

### 3.4. Characterization of Synthesized Molecules

The structures of the final product were characterized by 1D and 2D-NMR. The ^1^H NMR spectra of 12BzRoy-Sq, Roy-OA, and Roy-Sq were recorded on a Bruker 300 MHz NMR spectrometer and ^13^C, HSQC, and HMBC at 100 MHz at room temperature on a Bruker^®^ Biospin Fourier spectrometer at the Faculty of Pharmacy of the University of Lisbon. The chemical shifts (δ, ppm) were reported relative to the residual solvent peak, CDCl_3_ (δ, ppm), (Aldrich 99.80%, <0.01% H_2_O), 7.26 [^1^H] and 77.16 ppm [^13^C]).

#### 3.4.1. 12BzRoy-Squalene Conjugate

Yellow solid, yield 20.4%. ^1^H NMR (300 MHz, CDCl_3_ δ: 8.01 (s, 1H, H-1′), 7.98 (s, 1H, H-2′), 7.75 (s, 2H, H-6′), 7.72 (s, 1H, H-4′), 5.30 (s, 1H, H-7β), 4.28–4.16 (m, 2H, 6β, Sq) *, 3.48 (d, J = 7.1 Hz, 1H, Sq), 3.36 (sept, J = 7.1 Hz, 1H, H-15), 3.25 (t, J = 6.4 Hz, 2H, Sq), 2.19 (d, J = 1.9 Hz, 1H, Sq), 2.17 (s, 1H, Sq), 1.89–1.78 (m, 2H, H-2β, H-1β) *, 1.71–1.63 (m, 2H, H-2α, H-3) *, 1.60 (s, 10H, Sq) *, 1.33 (s, 1H, H-5), 1.31 (s, 10H, Sq) *, 1.29 (s, 3H, Me-17), 1.28 (s, 2H, Me-16), 1.27 (s, 3H, Me-19), 1.25 (s, 10H, Sq) *, 0.95 (s, 3H, Me-18) *, 0.07 (s, 3H). * Overlapped signals.

^13^C NMR (101 MHz, CDCl_3_) δ 184.32 (C-12), 183.40 (OCO-Bz), 153.20 (C-2′), 152.99 (C-14), 152.59 (C-6′), 146.65 (C-13), 140.67 (C-9), 138.61 (C-8), 136.31 (Sq), 133.21 (-4′), 126.25 (6OCO-Sq), 126.08 (C-5′), 125.81 (C-3′), 121.55 (Sq), 114.22 (-CO-Sq), 61.40 (C-7), 37.77 (C-18), 37.68 (Sq), 34.63 (C-4), 31.92 (Sq), 31.84 (C-3), 31.58 (Sq), 31.33 (Sq), 30.32 (Sq), 30.07 (Sq), 29.85 (Sq), 24.19 (C-15), 22.65 (C-19), 19.77 (C-17), 19.58 (C-16).

#### 3.4.2. Roy-Oleic Acid Conjugate

Yellow solid, yield 90.9%. ^1^H NMR (400 MHz, CDCl_3_) δ 5.64 (s, 1H, H-7β), 5.35 (s, 1H, OA-CH=CH), 4.31 (s, 1H, H-6α), 3.17 (Sept, 1H, H-15), 2.60 (q, J = 7.0, 6.5 Hz, 2H, OCO(CH_2_CH_2_, OA) *, 2.50 -2.59 (m, J = 11.7 Hz, H-2α, H-1β), 2.31–2.33 (d, J = 7.6 Hz, OCO(CH_2_), 2.19 (1H, s, OH-6β), 2.09–1.95 (m, 8H, OCOCH_3_, H-3α) *, 1.77 (s, 1H, H-2β), 1.62 (s, 3H, Me-20), 1.35 (H-5α), 1.33 (s, 11H, OA)*, 1.23–1.20 (m, 12H, Me-17, Me-16, Me-19) *, 1.19 (s, 3H, OA), 0.94 (s, 3H, Me-19), 0.88 (s, 3H, Me-OA). * Overlapped signals.

^13^C NMR (101 MHz, CDCl_3_) δ 185.39 (C-14),175.95 (C-11), 171.11 (OCO(CH2)n), 169.70 (OCOCH3), 152.93 (C-9), 149.75 (C-12), 139.43 (C-13), 135.65 (C-8), 68.89 (C-7), 67.39 (C-6), 49.84 (C-5), 42.34 (C-4), 39.05 (C-10), 33.88, (C-1), 33.59, (C-18), 31.35 (C-3), 27.31 (C-Oleic acid), 25.23 (C-15), 24.48 (C-2), 22.82 (C-19), 21.70 (C-20), 20.64 (C-16), 20.51 (OCOCH_3_), 20.23 (C-17).

### 3.5. Preparation of Self-Assembled Nanoparticles 

Self-assembled nanoparticles were prepared by the solvent displacement method. The drug-linker conjugate was dissolved in appropriate amount acetone to give a yellow solution of 4.0 mM in the drug. Nanoprecipitation was then induced by adding the solution dropwise and under stirring (400 rpm) to Milli-Q water (Sigma), without surfactant addition to a final concentration of 2.0 mM. The organic solvent was removed under reduced pressure, at 313 K. The turbid yellow suspension containing the nanoassemblies was stored in the dark, at 4 °C.

### 3.6. Physical-Chemical and Morphological Characterization of Self-Assembly Nanoparticles

#### 3.6.1. Fourier Transform Infrared Spectroscopy (FTIR) 

FTIR of derivatives **5** and **7** were evaluated by FTIR in a PerkinElmer^®^ Spectrum 400 (PerkinElmer Inc, Waltham, MA, USA) equipped with an attenuated total reflectance (ATR) device. The ATR system was cleaned before each analysis by using dry paper and scrubbing it with methanol and water (50:50). The room air FTIR-ATR spectrum was used as background to verify the cleanliness and to evaluate the instrumental conditions and room interferences due to H_2_O and CO_2_. The spectra were obtained collecting 100 scans of each sample, between 4000 and 600 cm^−1^, with a resolution of 4 cm^−1^. 

#### 3.6.2. Dynamic Light Scattering (DLS)

Physical characterization of the nanoparticles in suspension was carried out by evaluation of mean particle size, polydispersity index (PdI), and surface charge (zeta potential) by DLS and electrophoretic mobility in a Malvern Zetasizer Nano S (Malvern Instruments, Worcestershire, UK). Experiments were conducted in triplicate. Results were expressed as the mean ± SD (*n* = 3).

#### 3.6.3. Scanning Electron Microscopy (SEM) Analysis

The morphology of the self-assembled nanoparticles was analyzed with a JEOL-JSM7001F scanning electron microscope (SEM) at a voltage of 25 kV. The conductivity of the NP particle assemblies was increased by adding a thin coating of conductive gold/palladium (Polaron E-5100).

### 3.7. In Vitro Release Studies

The release pattern of Roy from Roy-OA NP was studied in vitro. Briefly, the self-assembled NPs were dissolved in 30 mL of PBS at approximately blood pH (pH 7.4, European Pharmacopoeia 7.0) under constant stirring (200 rpm) using a multi-position magnetic stirrer with heater RT series, 15, RT 15, IKA, to simulate the in vivo conditions [24]. Although the Roy-OA conjugate could self-assemble into NPs in Milli-Q water without any surfactant, this nanoassembly showed poor stability in PBS due to the highly hydrophobic surface. To address this issue, a small amount of Tween 80, 100 µL (Sigma-Aldrich), was added to prepare NPs for improved solubility. At appropriate time intervals, aliquots of the release medium were collected from three different points of the dissolution medium, to ensure a homogenous collection of the sample. Nanoparticles were isolated from the supernatant by centrifugation (7000× *g* rpm for 7 min). The amount of Roy collected from the in vitro release medium, at each time point, was determined by HPLC (Agilent Technologies 1260 Infinity II Series system with diode array detector (DAD; Agilent, Santa Clara, CA, USA and ChemStation Software (Hewlett-Packard, Alto Palo, CA, USA). 

The mobile phase consisted of a mixture of methanol (A), acetonitrile (B), and 0.3% trifluoroacetic acid in water (C) used as follows: 0 min, 15% A, 5% B, and 80% C; 2 min, 70% A, 30% B, and 0% C; 10 min, 70% A, 30% B, and 0% C; and 15 min, 15% A, 5% B, and 80% C. The time of analysis was 15 min, including the stabilization of the RP-18 column (Eclipse XDB-C18, 250 × 4.0 mm i.d., 5 µm) column, from Merck. The injection volume was 20 µL, the flow rate was set at 1 mL/min, and the detection wavelength was 270 nm. Roy identification was based on the comparison of retention time and ultraviolet (UV) spectra overlay with authentic standards. The assay was conducted for 72 h (*n* = 3, mean ± SD).

### 3.8. Preliminary Toxicity Assay

The Brine shrimp lethality assay (BSLA) is a benchtop assay used for the preliminary toxicity evaluation. The experiment was carried out using the method described by Epole et al., 2020 [25]. Brine shrimp eggs were obtained from JBL GmbH and Co. KG (Neuhofen, Germany). Briefly, *Artemia salina* cysts (brine shrimp eggs) were allowed to hatch in natural seawater, containing 35 g/L salts under constant aeration and illumination to ensure survival and maturity before use. Ten to fifteen nauplii were collected with the aid of a pipette and added to a 24-well plate. Drug conjugates and NPs were added to the corresponding wells, and the plate was stored for 24 h at 25 °C. Tests were carried out in triplicate. The negative control consisted of ten to fifteen nauplii per well in seawater without test samples, while K_2_Cr_2_O_7_ was used as the positive control. After the 24 h incubation, the number of dead nauplii in each well was recorded using a microscope. Larvae were considered dead after 5 s of no movement. Death was indued on the rest of the nauplii using K_2_Cr_2_O_7_ at 1 mg/mL, and the total nauplii in each was well determined. After 24 h, all dead larvae were counted, and the mortality rate (%) was determined according to Equation (1). Results were analyzed using GraphPad Prism version 5.00 for Windows, GraphPad Software, San Diego, CA, USA, www.graphpad.com, accessed on 5 February 2021. The results were expressed as the mean value ± SD, and a probability level *p* < 0.05 was considered to indicate statistical significance.
(1)$$\mathrm{Mortality}\;\mathrm{Rate}\;(\%)\;\frac{Total\;A.\;salina-Living\;A.\;salina}{Total\;A.\;salina}\;\times \;100$$

### 3.9. Cytotoxicity Study

#### 3.9.1. Cell Culture

The Vero-E6 cells line, representative of nontumor-like kidney cells, was obtained from ATCC (Manassas, VA, USA). Cells were cultured in DMEM supplemented with 10% fetal bovine serum, 100 U/mL penicillin, and 0.1 mg/mL streptomycin. The cultures were maintained at 37 °C under a humidified atmosphere containing 5% CO_2_. 

#### 3.9.2. Cell Viability

Cell Viability was determined by the MTT assay and is based on the reduction of 3-(4,5-dimethyl-2-thizolyl)-2,5-diphenyl-2H-tetrazolium bromide into formazan dye by active mitochondria of living cells. The effect of Roy and its NPs on cell viability was assessed. Briefly, 6.0 × 10^3^ Vero-E6 cells were cultured in 190 µL of complete medium in 96-well plates. Cells were grown for 24 h at 37 °C, 5 % CO_2_, and then exposed to different concentrations of Roy (13.75, 27.5 and 55 µM) and NPs (0.05, 0.1 and 0.2 µM) for 24 h. MTT reduction assay was performed as previously described by Fernandes et al. [26]. Each condition was tested in four replicates of two independent experiments.

### 3.10. Statistical Analysis

The results were expressed as the mean value ± SD. Comparisons were performed within groups using the Student’s *t*-test. Significant differences between control and experimental groups were assessed using GraphPad Prism version 5.00 for Windows, GraphPad Software, San Diego, CA, USA, www.graphpad.com, accessed on 5 February 2021. A probability level ** *p* < 0.01 was considered to indicate statistical significance.

## 4. Conclusions

In this study, the cytotoxic diterpenoids Roy, 12BzRoy, and DHR were explored as lead molecules for the hemi-synthesis of drug-linker conjugates. Using squalene, oleic acid, and 1-bromododecane as inducers, derivatives **5**, **7**, and **9** from squalene and oleic acid linkers were successfully prepared and confirmed by 1D- and 2D-NMR. Products **4** and **8** were synthesized; however, they were highly unstable. Nano assemblies of **5** and **9** were successfully characterized, and the cytotoxicity of **5** NPs was studied. Compound **5** and its corresponding NPs were tested against the nontumor-like kidney cell lines. The results revealed that **5**-NPs may be used as a strategy to deliver Roy, thereby increasing the water solubility using it as a suspension of self-assembly nanoparticles of Roy. The in vitro release profile of Roy in 5-NPs showed a delay on the release, which can be important issue for a potential therapeutic application of Roy.

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
