# Peer review of "Self-Assembly Nanoparticles of Natural Bioactive Abietane Diterpenes"

_ijms, 2021, doi:10.3390/ijms221910210_

Round 1

Reviewer 1 Report

Authors provided a paper entitled “Self-assembly nanoparticles of natural bioactive abietane diterpenes” for the publication on Processes, MDPI.

This paper has a good scientific soundness and deserves to be published after major revisions.

Here is the list of my issues:

Abietane is a diterpene that forms the structural basis for a variety of natural chemical compounds, for the plant extract used, the composition in diterpenes is not evaluated.Also, the study would have needed to be performed compared to a reference substance. 2.3. In vitro release studiesDrug release studies have clear regulations, please specify which equipment was used, what was the sampling protocol. Because they are bioactive drug compounds, the conditions chosen for in vitro release should be justified. 

3.0. Materials and methods

I recommend that the extraction method be described briefly.

The origin of the reagents used must be mentioned.

3.5. Preparation of self-assembled nanoparticles

The expression of temperature in ºC should be used. 

Author Response

Reviewer 1 corrections in Purple

Authors provided a paper entitled “Self-assembly nanoparticles of natural bioactive abietane diterpenes” for the publication on Processes, MDPI.

This paper has a good scientific soundness and deserves to be published after major revisions.

Here is the list of my issues:

1)  Abietane is a diterpene that forms the structural basis for a variety of natural chemical compounds, for the plant extract used, the composition in diterpenes is not evaluated.

We thank the reviewer for the comment. The composition of diterpenes in the plant extract used is beyond the scope of this work. Besides, this is done in previous works from our grup where P. madagascariensis and P. hadiensis phytochemical studies were done. See the following references that are also in the references of this work.

  1. Matias, D.; Nicolai, M.; Saraiva, L.; Pinheiro, R.; Faustino, C.; Diaz Lanza, A.; Pinto Reis, C.; Stankovic, T.; Dinic, J.; Pesic, M.; et al. Cytotoxic Activity of Royleanone Diterpenes from Plectranthus madagascariensis Benth. ACS Omega 2019, 4, 8094–8103, doi:10.1021/acsomega.9b00512.
  2. Ntungwe, E.; Mar, E.; Teod, C.; Teixid, S.; Capote, N.A.; Saraiva, L.; Mar, A. Preliminary Biological Activity Screening of Plectranthus spp . Extracts for the Search of Anticancer Lead Molecules. Pharmaceuticals 2021, 1–11.

2) Also, the study would have needed to be performed compared to a reference substance. 2.3. In vitro release studies Drug release studies have clear regulations, please specify which equipment was used, what was the sampling protocol. Because they are bioactive drug compounds, the conditions chosen for in vitro release should be justified.

We thank the reviewer for the comment. The equipment used and sampling protocol with reference are included in the manuscript.

3.0. Materials and methods

3) I recommend that the extraction method be described briefly.

The extraction method was briefly described.

4) The origin of the reagents used must be mentioned.

The origin of the reagents used are added in the manuscript.

3.5. Preparation of self-assembled nanoparticles

5) The expression of temperature in ºC should be used.

We thank the reviewer for the suggestion. The temperature was expression in °C

Reviewer 2 Report

Dear Authors

The manuscript entitled « Self-assembly nanoparticles of natural bioactive abietane diterpenes » submitted to International Journal of Molecular Sciences (MDPI) is well written, has important scientific message, and should be of great interest to the readers within th scope of nanotechnology, natural products synthesis, secondary metabolites derived from medicinal plants, and Anticancer activity of nanoparticules.

In this study, the auhtors described the synthesis of lipid-drug conjugates based on three abietane diterpenes (Roy, its derivative 12BzRoy, and DHR) which self-assemble in water in a way to increase the bioavailability. The nanoparticles of these lipid-drug conjugates were characterized and their properties as well as their cytotoxicity in different cancer cell lines were done.

- Abstract is comprehensive by itself. The important and essential information of the article is included.

- Structure and length: the overall structure of the article is well organized and well balanced. The article is written with the minimum length necessary for all relevant information.

- The introductory section adequately explains the framework and problems of the research

- Figures and tables are essential and clearly presented.

- The conclusion is logically supported by the obtained results.

- The analysis is statistically valid and follow the norms of the field.

- Novelty and originality: the article is novel and original. The article contains material that is new or adds significantly to knowledge already published in the field of nanotechnology and the pharmacology effects of nanoparticules derived from natural products.

The manuscript presents interesting and scientific important results related to nanoparticules. A few issues, however, need to be addressed:

** Abstract section : Line 30-32

The first molecule is the cytotoxic royleanone 7α-acetoxy-6β-hydroxyroyleanone (Roy, 1) that has been isolated from P. hadiensis (Forssk.) Schweinf. ex Sprenger leaves in large amount.

Please give the common name of the plant first, then the scientific name between bracket. For example, hairy coleus (Plectranthus hadiensis (Forssk.) Schweinf. ex Sprenger).

** Line 34 : isolated from the essential oil of P. madagascariensis

Same comment as mentioned before

** Line 40: Bioactivity of DHR.sq and Roy-OA NPs were lower when compared with Roy 40 and DHR,respectively

Please give or insert the IC50 of the cytotoxicity test for these conjugated compounds

** It is necessary to add or insert a list of abbreviation before the introduction section.

--- Introduction section

** Line 49: and effectively overcome some pharmacokinetic limitations [2]

Please could you exampling this paragraph by giving some examples of these limitations.

** Line 63: « interactions. . Thus »

Delete the extra comma.

** Line 63 : Thus, it is possible to obtain 63 fine and stable emulsion

Do you mean “nanoémulsion” instead of emulsion ?

** Line 86: royleanones

Please add some details about this secondary metabolite isolated from Plectranthus plants.

** Line 96: and has been reported with antioxidant, antimicrobial, and cytotoxic activities

Please add some references at the end of this paragraph.

** Line 101 : You need to add a reference after this paragraph.

** Line 208: 3.0. Materials and Methods 3.1. Plant Material

What about the identification of the plants used in this study ? the voucher specimen number ? The harvesting period ? the quantity used ? the floral stage of the plant? All of these information must be added in the plant material section.

Also, it is necessary to add an image describing the plant material harvested.

** Line 214 : The extraction and isolation processes of 1 and 3 were performed according to Ntungwe 214 et al., 2021 [21], and Garcia C. et al., 2018 [19] respectively.

It is necessary that authors give briefly the extraction procedure method used for isolation of interesting compounds.

** Line 324 : 3.7. In vitro release studies

I can not find a reference of the technique used for the release pattern of Roy from Roy-OA NP.

Please could you insert a suitable reference.

** Line 370 : L-glutamine instead of l-glutamine

** Line 391: Briefly, 6.0 x 103 Vero-E6 cells

Do you mean 6.0 x 103 ?

---- General comments :

Poor Writing and Organization

- Inadequate description of methods

- Discussion that only repeats the results but does not interpret them. For example,

** Line 183 : 2.4.1. Preliminary toxicity assay

** Line 192: 2.4.2. Cytotoxicity study

Lack of interpretations / Discussion need to be detailed. The authors should have a sufficient know-how to interpret the exact reasons of the research outcome. This article does not  provide sufficient information and in-depth discussion.

Final decision: Accept with major revisions

Author Response

 Reviewer 2 Corrections in Green

Dear Authors

The manuscript entitled « Self-assembly nanoparticles of natural bioactive abietane diterpenes » submitted to International Journal of Molecular Sciences (MDPI) is well written, has important scientific message, and should be of great interest to the readers within the scope of nanotechnology, natural products synthesis, secondary metabolites derived from medicinal plants, and Anticancer activity of nanoparticules.

In this study, the authors described the synthesis of lipid-drug conjugates based on three abietane diterpenes (Roy, its derivative 12BzRoy, and DHR) which self-assemble in water in a way to increase the bioavailability. The nanoparticles of these lipid-drug conjugates were characterized and their properties as well as their cytotoxicity in different cancer cell lines were done.

- Abstract is comprehensive by itself. The important and essential information of the article is included.

- Structure and length: the overall structure of the article is well organized and well balanced. The article is written with the minimum length necessary for all relevant information.

- The introductory section adequately explains the framework and problems of the research

- Figures and tables are essential and clearly presented.

- The conclusion is logically supported by the obtained results.

- The analysis is statistically valid and follow the norms of the field.

- Novelty and originality: the article is novel and original. The article contains material that is new or adds significantly to knowledge already published in the field of nanotechnology and the pharmacology effects of nanoparticules derived from natural products.

The manuscript presents interesting and scientific important results related to nanoparticules. A few issues, however, need to be addressed:

** Abstract section : Line 30-32

The first molecule is the cytotoxic royleanone 7α-acetoxy-6β-hydroxyroyleanone (Roy, 1) that has been isolated from P. hadiensis (Forssk.) Schweinf. ex Sprenger leaves in large amount.

Answers from the Authors:

Please give the common name of the plant first, then the scientific name between bracket. For example, hairy coleus (Plectranthus hadiensis (Forssk.) Schweinf. ex Sprenger).

We thank the reviewer for the correction. The common name of Plectranthus hadiensis (hairy coleus (Forssk.) Schweinf. ex Sprenger was included.

** Line 34 : isolated from the essential oil of P. madagascariensis.

Same comment as mentioned before

The common name of Plectranthus  madagascariensis (thicket coleus) was included.

** Line 40: Bioactivity of DHR.sq and Roy-OA NPs were lower when compared with Roy and DHR,respectively

Please give or insert the IC50 of the cytotoxicity test for these conjugated compounds

We thank the reviewer for the suggestion. The cytotoxicity of DHR.sq and Roy-OA NPs was included in the manuscript.

** It is necessary to add or insert a list of abbreviation before the introduction section.

The abbreviations used in this work are all standard scientific abbreviations hence in our opinion there is no need to  redefining them.

--- Introduction section

** Line 49: and effectively overcome some pharmacokinetic limitations [2]

Please could you be exampling this paragraph by giving some examples of these limitations.

We thank the reviewer for the comment. Examples of pharmacokinetic limitations such as bioavailability of drug, immunogenicity, and possible side effects due to targeted delivery to specific tissues are included in the manuscript.

** Line 63: « interactions. . Thus »

Delete the extra comma.

The manuscript was corrected accordingly.

** Line 63 : Thus, it is possible to obtain 63 fine and stable emulsion

Do you mean “nanoémulsion” instead of emulsion ?

We thank the reviewer for the comment. Yes, that is correct, we mean Nanoemulsion. The manuscript was corrected accordingly.

** Line 86: royleanones

Please add some details about this secondary metabolite isolated from Plectranthus plants.

We thank the reviewer for the suggestion. More details on royleanones were added to the manuscript.

** Line 96: and has been reported with antioxidant, antimicrobial, and cytotoxic activities

Please add some references at the end of this paragraph.

We thank the reviewer for the suggestion. The refences were added in the manuscript.

** Line 101 : You need to add a reference after this paragraph.

We thank the reviewer for the suggestion. The refence was added in the manuscript.

** Line 208: 3.0. Materials and Methods 3.1. Plant Material

What about the identification of the plants used in this study ? the voucher specimen number ? The harvesting period ? the quantity used ? the floral stage of the plant? All of these information must be added in the plant material section.

We thank the reviewer for the suggestion. The plant used in this study was described in detail. The identification of the plants, voucher specimen number, harvesting period, floral stage and quantity used were added in the manuscript.

Also, it is necessary to add an image describing the plant material harvested.

We thank the reviewer for the suggestion. The image of the plant material harvested was added (Figure 9).

** Line 214 : The extraction and isolation processes of 1 and 3 were performed according to Ntungwe 214 et al., 2021 [21], and Garcia C. et al., 2018 [19] respectively. It is necessary that authors give briefly the extraction procedure method used for isolation of interesting compounds.

We thank the reviewer for the suggestion. The extraction procedure method used for isolation of interesting compounds is briefly described now.

** Line 324 : 3.7. In vitro release studies

I can not find a reference of the technique used for the release pattern of Roy from Roy-OA NP. Please could you insert a suitable reference.

We thank the reviewer for the suggestion. A suitable reference is added accordingly (Reference [26]).

** Line 370 : L-glutamine instead of l-glutamine

We thank the reviewer for the comment. The manuscript was corrected accordingly.

** Line 391: Briefly, 6.0 x 103 Vero-E6 cells. Do you mean 6.0 x 103 ?

We thank the reviewer for the comment. No, we mean 6.0 x 103. The manuscript was corrected accordingly.

---- General comments :

Poor Writing and Organization

We thank the reviewer for the comment. The manuscript was carefully edited and the Organization improved.

- Inadequate description of methods

We thank the reviewer for the comment. The methods were described in details and proper references made.

- Discussion that only repeats the results but does not interpret them. For example,

** Line 183 : 2.4.1. Preliminary toxicity assay

We thank the reviewer for the comment. The Preliminary toxicity results were properly interpreted.

** Line 192: 2.4.2. Cytotoxicity study

We thank the reviewer for the comment. The Cytotoxicity results were properly interpreted.

Lack of interpretations / Discussion need to be detailed. The authors should have a sufficient know-how to interpret the exact reasons of the research outcome. This article does not  provide sufficient information and in-depth discussion.

We thank the reviewer for the comment. The results were properly interpreted and discussed in more detail.

** After our careful check, we noticed that the order of sections is not correct in your manuscript. According to rules, the order of the sections of the article should

be adjusted to:

  1. Introduction
  2. Results
  3. Discussion
  4. Materials and Methods
  5. Conclusions

The manuscript was corrected accordingly.

Final decision: Accept with major revisions

Round 2

Reviewer 1 Report

I accept the publication of the article.

Reviewer 2 Report

Dear Author

After careful examination of your revised version of your article entitled « Self-assembly nanoparticles of natural bioactive abietane diterpenes » re-submitted to International Journal of Molecular Sciences (MDPI), we notice that the authors have responded to all of our questions with details.

The article is well written, has important scientific message, and should be of great interest to the readers within the scope of nanotechnology, natural products synthesis, secondary metabolites derived from medicinal plants, and Anticancer activity of nanoparticules. Also, this article is novel and original. It contains material that is new or adds significantly to knowledge already published in the field of nanotechnology and the pharmacology effects of nanoparticules derived from natural products.

Final decision : Accept.